# Dietary Copper and Selenium Intakes and the Risk of Type 2 Diabetes Mellitus: Findings from the China Health and Nutrition Survey

**DOI:** 10.3390/nu14102055

**Published:** 2022-05-13

**Authors:** Zhixin Cui, Haiyan Zhou, Ke Liu, Man Wu, Shun Li, Shuangli Meng, Huicui Meng

**Affiliations:** 1School of Public Health (Shenzhen), Shenzhen Campus of Sun Yat-sen University, Sun Yat-sen University, Shenzhen 518107, China; cuizhx3@mail2.sysu.edu.cn (Z.C.); zhouhy73@mail2.sysu.edu.cn (H.Z.); liuk227@mail2.sysu.edu.cn (K.L.); wuman5@mail2.sysu.edu.cn (M.W.); lish336@mail2.sysu.edu.cn (S.L.); mengshl@mail2.sysu.edu.cn (S.M.); 2Guangdong Provincial Key Laboratory of Food, Nutrition and Health, Guangzhou 510080, China; 3Guangdong Province Engineering Laboratory for Nutrition Translation, Guangzhou 510080, China

**Keywords:** dietary copper, dietary selenium, type 2 diabetes mellitus, Chinese adults

## Abstract

The long-term associations between dietary copper (Cu) and selenium (Se) intakes and type 2 diabetes mellitus (T2DM) risk are unclear. We aimed to examine the prospective associations between dietary Cu and Se intakes and T2DM risk in Chinese adults. A total of 14,711 adults from the China Health and Nutrition Survey (1997–2015) were included. Nutrient intakes were assessed by 3 consecutive 24 h recalls and food-weighing methods. T2DM was identified by a validated questionnaire and laboratory examination. Cox regression models were used for statistical analysis. A total of 1040 T2DM cases were diagnosed during 147,142 person-years of follow-up. In fully adjusted models, dietary Cu or Se intake was not associated with T2DM risk. Dietary Se intake significantly modified the association between dietary Cu intake and T2DM risk, and dietary Cu intake was positively associated with T2DM risk when Se intake was lower than the median (*p*-interaction = 0.0292). There were no significant effect modifications on the associations by age, sex, BMI, or region. Although dietary Cu or Se intake was not independently associated with T2DM risk in Chinese adults free from cardiometabolic diseases and cancer at the baseline, there was a significant interaction between dietary Cu and Se intakes on T2DM risk.

## 1. Introduction

Despite decades of efforts to reduce cardiometabolic risk, the prevalence of type 2 diabetes mellitus (T2DM), which continues to be a leading public health challenge in China, has been increasing dramatically in China over the last decades [1]. Adherence to healthy lifestyle behaviors, especially following healthy dietary patterns and eating behaviors, are key strategies for T2DM prevention [2]. The deficiency or excess of selenium (Se), copper (Cu), calcium, iron, manganese, and magnesium have been reported to contribute to insulin resistance and imbalances of glucose homeostasis [3,4].

Cu is an essential mineral serving as a catalytic cofactor of enzymes such as cytochrome c oxidase and Cu/Zn superoxide dismutase, and it acts as both a pro-oxidant and an antioxidant [5,6]. A previous review has reported that imbalance of Cu may promote the pathogenesis of diabetes mellitus by inducing elevated susceptibility to oxidative damage [4], and a meta-analysis has reported a positive association between serum Cu concentrations and the risk of T2DM [7]. However, studies on the associations between dietary Cu intake and the risk of T2DM are scarce, and no data have been reported for Chinese adults. There is only one cohort study in Japan that has reported a positive relationship between dietary Cu intake and T2DM risk [8]. As Japan has different dietary habits from China [9,10], whether the association between dietary Cu intake and the risk of T2DM in Japanese adults is identical to the association in Chinese adults is unclear. Therefore, it is necessary to explore the association between dietary Cu intake and the risk of T2DM in Chinese adults. In addition, data on this topic may contribute to the update of the Dietary Reference Intakes (DRIs) of Cu in China [11].

Se is also an essential mineral with a toxicological property when its intake level is excessive [12]. Animal studies have shown that both selenium deficiency and excess impair pancreatic β-cell disfunction and contribute to the pathogenesis of insulin resistance [12,13,14,15], which suggests that optimal dietary Se intake might be essential for T2DM prevention. Some meta-analyses and cohort studies in Italy have shown that increased dietary Se intake is associated with higher risk of T2DM [16,17,18,19]. In China, a previous cross-sectional study has reported positive association between dietary Se intake and prevalence of diabetes in middle-aged and older adults in Hunan province [20]. In addition, another recent cross-sectional study conducted in North China has found identical results, in which higher dietary Se intake is associated with higher risk of T2DM [21]. In order to explore the long-term associations, prospective cohort studies are needed to confirm the relationship between dietary Se intake and the incidence risk of T2DM in China.

The aim of the present study was to assess the associations between dietary Cu and Se intakes and risk for T2DM in a nationwide cohort of Chinese adults. We hypothesized that higher dietary Cu and Se intakes were associated with an increased risk for T2DM.

## 2. Materials and Methods

### 2.1. Study Population

The China Health and Nutrition Survey (CHNS) is an ongoing international collaborative longitudinal study to evaluate the impact of social and economic transformation on the health and nutritional status of Chinese residents. The CHNS was initiated in 1989, and 10 rounds of surveys were completed by 2015. The survey used a multistage random cluster process to recruit 7200 families and more than 30,000 participants from 15 provinces, with great differences in geography and economy. The details of the purpose and design of CHNS have been described previously [22,23]. The study was in accordance with the Declaration of Helsinki guidelines. All study procedures were approved by the Institutional Review Committees of the University of North Carolina at Chapel Hill (Project identification code 07-1963) and the National Institute for Nutrition and Health at the Chinese Center for Disease Control and Prevention (Project identification code 201524) [24,25], and informed consent forms were signed by all participants prior to the survey [23].

The current prospective cohort study was conducted with the use of the available data from CHNS, wave 1997 to 2015, and the baseline for each participant was defined as the date of their first participation in the dietary survey during this period. In the current investigation, participants below 18 years old (*n* = 8992) at baseline, with missing values in all dietary (*n* = 3622) or physical examination records (*n* = 98), or who were diagnosed with T2DM or other diseases (such as myocardial infarction, stroke, or tumor) that may alter dietary intake and eating behaviors or who were taking medicine known to affect lipid or glucose metabolism at baseline (*n* = 849) were excluded. In addition, participants with an implausible total energy intake (<800 or >4200 kcal/day for men, <500 or >3500 kcal/day for women, *n* = 677) at baseline [26], who were pregnant or breast-feeding at baseline (*n* = 902), or who participated in the survey for only one round (*n* = 3463) were further excluded. A total of 14,711 participants (7333 men and 7378 women) were included in the final analysis (see Figure 1 for participant flow).

### 2.2. Dietary Assessment

Dietary intake data of participants were collected with the use of three consecutive 24 h recalls at the individual level [27]. Each participant was asked by trained interviewers about the type, amount, preparation method, and dining location of every single food and beverage consumed within the past 24 h for three consecutive days, including two working days and one weekend [27]. The amount of food consumption was confirmed by a food weighing method at the household level during the same 3-day period [28].

The daily intake of a certain nutrient was calculated by multiplying the edible portion by the nutrient content of each food and then summing up the nutrient intake of all foods. The nutrient contents and energy of each food were obtained from the China Food Composition Tables by food codes [29,30,31,32]. In the current analyses, the cumulative average intakes of dietary total energy, Cu, Se, total protein, animal protein, plant protein, fat, saturated fatty acids (SFA), monounsaturated fatty acids (MUFA), polyunsaturated fatty acids (PUFA), cholesterol, carbohydrate, and fiber from baseline to the last date before diagnosis of T2DM, death, or the end of follow-up were calculated to represent the long-term dietary nutrient intakes. All nutrient intakes were adjusted for total energy by the regression residual method [33], and macronutrients were expressed as the percentage of total energy (%E). The food sources of dietary Cu and Se intakes of the participants were categorized into “grains and tubers”, “legumes”, “vegetables, fruits, fungi, and algae”, “meat”, “eggs”, “fish and seafoods”, “condiments”, “ethnic foods, cakes, and fast food”, and “others”. Dietary glycemic index (GI) values for each participant were calculated based on a previously established method [34], with the amount of available carbohydrate of carbohydrate-containing food items and their corresponding GI values obtained from either the China Food Composition Table [32] or the International Tables of Glycemic Index and Glycemic Load Values 2008 [35]. If there was a lack of time for the dietary survey, the questionnaire survey time of the same survey year would prevail.

### 2.3. Assessment of T2DM

The risk of new-onset T2DM occurring between 2000 and 2015 in the CHNS cohort was the primary outcome of the current analysis. The definition of T2DM in the current study was consistent with previous studies assessing the associations between diet and new-onset T2DM risk using data from CHNS cohort [36,37]. Briefly, information on the health status and medical history of the participants was collected by professional interviewers via questionnaires during each wave of follow-up. A set of three questions were asked to confirm whether the participants had ever had a diagnosis of diabetes: (1) Did the physician give you a diagnosis of diabetes? (2) If yes, how old were you when the physician gave you the diagnosis? (3) Have you ever used the following treatments for glucose control or diabetes, including special diet, weight control, oral medicine, insulin injection, traditional Chinese medicine, folk prescription, qi-gong, and other methods? [38]. If the participant answered yes to any of the three questions, they were confirmed as diabetic patients [37]. In the 2009 wave, fasting concentrations of blood glucose and glycosylated hemoglobin (HbA1c) were assessed, and these data were used as supplementary criteria for the confirmation of diabetes. Participants with a fasting blood glucose concentration ≥ 7 mmol/L or HbA1c ≥ 6.5% were defined as T2DM cases [37]. Since participants included in the final analysis were at least 18 years old, diagnosis of type 1 diabetes mellitus was largely ruled out, due to its early onset and low prevalence rate in China. In addition, participants who were pregnant at baseline were excluded in the analysis, and thus, the diagnosed DM cases were basically T2DM, being exclusive of gestational diabetes mellitus [39,40]. If there were inconsistent records in the different waves of follow-up, the first wave record would prevail [37].

### 2.4. Assessment of Other Covariates

The sociodemographic and lifestyle information of participants, including sex, age, province of residence, educational level, smoking status and alcohol consumption, were collected by trained interviewers through validated questionnaires [23]. Educational levels were categorized into three categories, including primary (primary school or lower), middle (middle school), or high (high school or above). Smokers and alcohol drinkers were defined as having smoked or drunk alcohol since baseline to last wave before the end of follow up. The body height, weight, waist, and hip circumferences and blood pressures of participants were measured in accordance with standard methods by using calibrated instruments [41]. Body mass index (BMI) was calculated as weight (kg) divided by the square of height (m^2^) [42]. CHNS comprehensively evaluated the economic activity, social services, population density, education, housing, sanitation, communications, traditional markets, modern markets, transportation infrastructure, health infrastructure, and diversity, and reflected the urbanicity of the participants’ community of residence with a scale score of 1 to 120 on the urbanization index [23]. In the current analysis, the urbanization index was classified in tertiles as low (23.0–49.6), moderate (49.6–75.0), or high (75.0–103.0). According to the geographical Qinling–Huaihe line, the provinces were divided into northern or southern regions [43]. The northern regions included Beijing, Liaoning, Heilongjiang, Shandong, and Henan, and the southern regions included Shanghai, Jiangsu, Hubei, Hunan, Guangxi, Guizhou, and Chongqing. Physical activity status was evaluated based on the time and intensity of occupational, household, leisure-time, and transportational activities, and expressed as metabolic equivalent-hours/week (MET-h/wk) [44].

### 2.5. Statistical Analysis

All statistical analyses were performed with SAS 9.4 (SAS Institute, Cary, NC, USA) and R software (version 3.6.3, The R Foundation for Statistical Computing, Vienna, Austria), and a two-tailed *p* < 0.05 was considered as statistically significant. Sociodemographic, anthropometric, and lifestyle characteristics of the participants at baseline were presented according to quintiles (Q) of energy adjusted dietary Cu and Se intakes. Continuous variables were presented as mean ± standard deviation (SD), and categorical variables were presented as *n* (%).

The baseline time of each participant was defined as the date of their first participation, with complete dietary data since 1997, and the individual follow-up person-years of each participant were calculated from the baseline to the date of first diagnosis with T2DM, death, the last wave before departure from the survey, or the end of the last survey at 2015, whichever came first [37,45]. In order to reduce inter-individual variation and capture long-term dietary patterns, cumulative average values of dietary intake, BMI, urbanization index, and physical activity status data were calculated and used in the analysis [46]. These data were not updated for the cumulative average calculation following diagnosis with T2DM to minimize potential confounding attributed to changes in dietary intake or lifestyle behaviors after diagnosis with chronic diseases.

Time-dependent Cox proportional hazard regression was performed to investigate the associations between dietary Cu or Se intakes and the risk of T2DM [45]. The independent variables were quintiles of dietary Cu or Se intakes, and the lowest quintile was used as the reference group to estimate hazard ratios (HRs) and 95% confidence intervals (95% CI). The median of each quintile was used as a continuous variable in the Cox regression model in the test for linear trend analysis. Model 1 adjusted for sex (female or male) and age (<50, 50–54, 55–59, 60–64, or ≥65 years). Model 2 further adjusted for sociodemographic, lifestyle, and dietary confounders, including BMI [underweight (<18.5 kg/m^2^), normal weight (18.5–23.9 kg/m^2^), overweight (24–27.9 kg/m^2^), or obese (≥28 kg/m^2^)], urbanization index [tertiles as low (23.0–49.6), moderate (49.6–75.0), or high (75.0–103.0)], region [northern (Beijing, Liaoning, Heilongjiang, Shandong, and Henan) or southern (Shanghai, Jiangsu, Hubei, Hunan, Guangxi, Guizhou and Chongqing)], educational level [primary (primary school or lower), middle (middle school), or high (high school or above)], alcohol intake (ever drinking or not), smoking (ever smoking or not), physical activity status (tertiles), baseline hypertension [no (SBP < 140 mm Hg and DBP < 90 mm Hg) or yes (SBP ≥ 140 mm Hg or DBP ≥ 90 mm Hg)], and dietary variables, including total energy intake, cholesterol, fiber, dietary GI, ratio of animal protein to plant protein, and ratio of PUFA to SFA (all in quintiles). Stratified analysis and potential effect modification were tested for the associations between dietary Cu and Se intakes and the risk of T2DM on the basis of sex (female or male), age (<60 or ≥60 years old), BMI (<24 or ≥24 kg/m^2^), region (northern or southern), or baseline hypertension history (baseline hypertension or with normal blood pressure). Joint analysis was performed to test the potential interactions of dietary Cu and Se intakes on the risk of T2DM.

## 3. Results

### 3.1. Sociodemographic, Anthropometric and Lifestyle Characteristics of Study Participants at Baseline

Data from a total of 14,711 participants were analyzed in this cohort study and 1040 new-onset T2DM cases were documented during the median follow-up duration of 9.2 years (147,142 person-years) from 1997 to 2015. Distribution of sociodemographic, anthropometric, and lifestyle characteristics of the participants at baseline according to quintiles (Q) of energy-adjusted Cu and Se intakes is presented in Table 1. The mean age of the participants was 44 ± 15 years, 49.8% were males, and the mean BMI was 23.2 ± 3.3 kg/m^2^. The mean daily intakes of Cu and Se were 1.9 ± 0.6 mg/day and 41.4 ± 17.6 µg/day, respectively (Table 1). The percentage of male participants, smoking status, dietary intakes of cholesterol, and dietary GI values of the participants were similar across the quintiles of Cu intakes (Table 1). In general, participants with higher intakes of Cu were more likely to be younger, have lower BMI, reside in areas with a low urban index and in northern provinces, be less educated, to drink alcohol, be physically active, and have higher dietary intakes of Se, total energy, protein, plant protein, PUFA: SFA ratio, carbohydrate and fiber, and lower intakes of animal protein, animal protein: plant protein ratio and dietary fat, SFA, MUFA, and PUFA in comparison to those who had lower Cu intakes (Table 1). The proportion of smokers and the average dietary GI values of the participants were similar across the quintiles of Se intakes. Participants with higher intakes of Se were more likely to be younger, males, have higher BMI, reside in areas with a high urban index and in northern provinces, have higher education levels, drink alcohol, be physically inactive, have hypertension, and have higher intakes of dietary protein, animal protein, animal protein: plant protein ratio and dietary fat, SFA, MUFA, PUFA, cholesterol and fiber, and lower intakes of dietary total energy, plant protein, carbohydrate, and PUFA: SFA ratio in comparison to those who had lower Se intakes (Table 1).

### 3.2. Food Sources of Dietary Cu and Se Intakes

The top three food sources of dietary Cu intakes were food groups of grains and tubers, vegetables, fruits, fungi, algae and legumes (Table 2). The main food sources of dietary Se intakes were grains and tubers, following by meat and eggs (Table 2).

### 3.3. Associations between Dietary Cu and Se Intakes and Risk of T2DM

In Cox regression models adjusted for sex and age (Model 1), dietary Cu intake was not significantly associated with the risk of T2DM (*p*-trend = 0.17), while dietary Se intake was positively associated with the risk of T2DM (*p*-trend < 0.0001) (Table 3). In fully adjusted Cox regression models (Model 2), although there was a significant association between dietary Cu intake and the risk of T2DM in the trend test (*p*-trend = 0.0130), none of the quintiles was significantly different compared to Q1 (all *p* > 0.05). Dietary Se intake was also not significantly associated with the risk of T2DM in the fully adjusted model (Model 2, *p*-trend = 0.82) (Table 3).

A significant interaction between dietary Cu and Se intake was observed. Dietary Se intake significantly modified the association between dietary Cu intakes and T2DM risk (*p*-interaction = 0.0292), and dietary Cu intake was positively associated with an increased T2DM risk when the intake of Se was lower than the median intake (Q5 vs. Q1: HR = 1.78, 95% CI: 1.20, 2.65; *p* = 0.0045) (Figure 2A). No significant effect modification was observed between dietary Se intake and T2DM risk by dietary Cu intakes (*p*-interaction = 0.0922) (Figure 2B).

### 3.4. Associations between Dietary Cu and Se Intakes and Risk of T2DM Based on Potential Effect Modifiers

There was no evidence of effect modification of the associations between dietary Cu and Se intakes and T2DM risk by age, sex, BMI, or region (all *p*-interaction > 0.05) (Table 4). Baseline hypertension history significantly modified the association between dietary Cu intake and T2DM risk (*p*-interaction = 0.0347). In the stratified analysis, although there was a significant association between dietary Cu intake and T2DM risk in participants without hypertension at baseline (*p*-trend = 0.0077), none of the quintiles was significantly different compared to Q1 (all *p* > 0.05).

## 4. Discussion

In Chinese adults free from cardiometabolic diseases and cancer at entry of a prospective cohort, dietary Cu or Se intakes were not significantly associated with T2DM risk during a median follow-up of 147,142 person-years. There was a significant interaction between dietary Cu and Se intakes on the risk of T2DM, and dietary Se intake significantly modified the association between dietary Cu intake and T2DM risk. To the best of our knowledge, this is the first prospective cohort study determining the association between dietary Cu and Se intakes and T2DM in Chinese adults, and we found, for the first time, a joint effect of dietary Cu and Se intakes on the risk of T2DM.

Previous studies on the association between dietary Cu and T2DM risk and related risk factors are limited. A cross-sectional study by Bo et al. [47] reported that dietary Cu intake is inversely associated with fasting glucose concentrations, and studies on Cu supplementation have also reported reduced glucose levels following supplementation [48]. Consistently, an animal experiment has also confirmed that Cu restriction may cause impaired insulin secretion [49]. These results suggest an elevated risk of T2DM associated with low intake of dietary Cu. The underlying mechanism responsible for the inverse relationship may be partly attributed to the impairment of the copper-dependent cytochrome c oxidase in the pancreatic islet by low copper intake [50], resulting in dysregulation of the function of pancreatic *β*-cells [51]. Since ceruloplasmin, together with transferrin, plays an important role in Fe transportation, low dietary Cu intake may also lead to disturbed Fe metabolism, which has been reported to related to T2DM risk [37,52]. However, the current study found no significant association between dietary Cu intake and the risk of T2DM. Prior to our study, there was only one epidemiological study on the association between dietary Cu intake and T2DM risk in Japan, which reported a positive association between dietary Cu intake and T2DM risk [8]. This finding is contrary to the result of the current study. The heterogenous results may be attributed to the differential dietary habits between Japanese and Chinese adults [9,10], which may play an important role in altering Cu absorption and bioavailability [53]. These data require further investigation in larger prospective cohort studies, and exploration of the causal relationships between dietary Cu intake and T2DM risk and related risk factors in randomized controlled trials in China are also required to provide useful information for updating the DRIs of Cu [11].

Consistent with the results observed for dietary Cu intake and T2DM risk, dietary Se intake was not significantly associated with T2DM risk. This finding is in contrast to most of the data from observational studies reported previously, as well as some recent meta-analyses, which have reported associations between higher dietary Se intakes and increased risk for T2DM [16,17,19,54]. Two cross-sectional studies among Chinese populations have also reported a positive association between dietary Se intake and risk of T2DM [20,21]. According to these previous studies, potential mechanisms underlying the positive association may be partially attributed to the adverse effects of high doses of Se compounds and species or the subsequent upregulation of selenoprotein biosynthesis on glucose homeostasis [12,17]. A recent case control study has reported that increased dietary Se intake is positively associated with a higher prevalence of T2DM, oxidative stress, and increased serum concentrations of proinflammatory cytokines concurrently, suggesting that oxidative stress and inflammation may mediate the relationship between high Se intake and T2DM risk [55]. In addition, positive associations between dietary Se intake and phylum *Synergistetes* and genus *Prevotella_7* have been observed in a small cohort of Chinese Uyghur adults, in which two thirds are patients with newly diagnosed T2DM or impaired glucose regulation [55]. An animal study has reported a higher relative abundance of *Porphyromonadaceae* and *Tanerella*, and lower relative abundance of *Alistipes* and *Parabacteroides* in mice fed a Se-enriched diet in comparison to a Se-deficient diet [56]. Considering the role of dysbiosis in T2DM development and related risk factors [57,58], alterations in gut microbiota composition and diversity may be another proposed mechanism responsible for the association between high Se intake and increased T2DM risk. Notwithstanding the consistent findings in earlier observational studies, our study did not confirm a long-term association between dietary Se intake and T2DM risk, and the data to support the causal effect of a high dose of Se intake via supplementation on T2DM risk have also been heterogenous. Among the five randomized controlled trials available for this topic, one study has reported an increased risk of T2DM following Se supplementation compared to placebo control [59], while other studies have reported null effect [60,61,62,63]. Differences in baseline selenium levels of participants and forms of Se supplementation may partially explain the discordant results [17,55]. These randomized controlled trials have been conducted in Western countries [17,55]. Lacking data in this area, further interventional studies assessing the effect of dietary Se or Se supplementation on the prevalence of T2DM and related risk factors in Chinese adults are required.

There are significant interactions of dietary Cu intake with dietary Se intake on the risk of T2DM in the current study. Dietary Se intake significantly modified the association between dietary Cu intake and T2DM risk, and the most elevated T2DM risk was observed in participants with both high dietary Cu and low dietary Se intake. Although no previous studies have reported a similar joint effect of dietary Cu and Se on T2DM risk, several studies have demonstrated links between Cu and Se. A randomized trial in low-birthweight infants has reported that serum Cu concentration decreases after adding Se supplementation to parenteral nutrition and increases when serum Se concentration decreases after parenteral nutrition is discontinued, which suggests that Se supplementation might affect Cu metabolism [64]. The study of Evans et al. [65] has reported that the Cu level in erythrocyte is associated with *SELENBP1* genes, which encodes selenium binding protein 1. Potential mechanisms responsible for the interactions between dietary Cu and Se intakes on T2DM risk require further investigation.

There are several strengths of the current study. The survey design was rigorous, and the questionnaires and physical and laboratory examinations were all conducted by trained staff under strict quality control, making the data collection reliable. No prior epidemiological study has determined the relationship between dietary Cu intake and T2DM risk in the Chinese population, and previous studies assessing the associations between dietary Se intake and T2DM risk in the Chinese population are limited to a cross-sectional design and people from a single province. Our study was able to investigate the long-term associations in Chinese adults from 15 provinces or mega-cities via using a prospective design with a long follow-up time. We also examined potential effect modifications for the associations between dietary Cu and Se intakes and risk of T2DM by sex, age, BMI, region, or baseline hypertension history, which were less explored previously. The quality of dietary intake data was guaranteed by using combinations of three consecutive 24 h dietary recall records at the individual level and a food-weighing method at the household level to further confirm the amount of consumption.

Some limitations of the study need to be addressed. Although potential confounding factors have been adjusted in the fully adjusted models, there might be other residual confounding factors that were not included, which is a similar issue in other observational studies. The 24 h dietary recall records may not be as good as food-frequency questionnaires in evaluating long-term dietary patterns and eating habits. However, this limitation may be partially overcome in our analysis by calculating the cumulative average dietary intake data as the exposure variables. Information on food material preparation, cooking methods, and cooking time were not included in the analysis. The confirmation of T2DM cases were based on self-reported questionnaires rather than clinical testing, which was difficult to perform in a large-scale cohort study. This limitation was partially overcome by using data for fasting glucose and HbA1c concentrations in the 2009 wave for ascertainment of T2DM cases. In addition, potential biological mechanisms were not explored due to the observational nature of this study.

## 5. Conclusions

In conclusion, we for the first time found no significant association between dietary Cu intake and the risk of T2DM in Chinese adults. Consistently, there was no association between dietary Se intake and T2DM risk. There was a significant interaction between dietary Cu and Se intakes. Dietary Se intake significantly modified the association between dietary Cu intake and T2DM risk, and high dietary Cu intake, together with low dietary Se intake, are more strongly associated with an elevated risk of T2DM. Our findings raised the intriguing possibility that optimal combinations of dietary Cu and Se intake levels may be important for preventing T2DM in Chinese adults. These findings add new information to the current literature, suggesting that there should be a reassessment of the relationships between dietary micronutrient intake and T2DM risk intended to aid in T2DM prevention.

## Figures and Tables

**Figure 1 nutrients-14-02055-f001:**
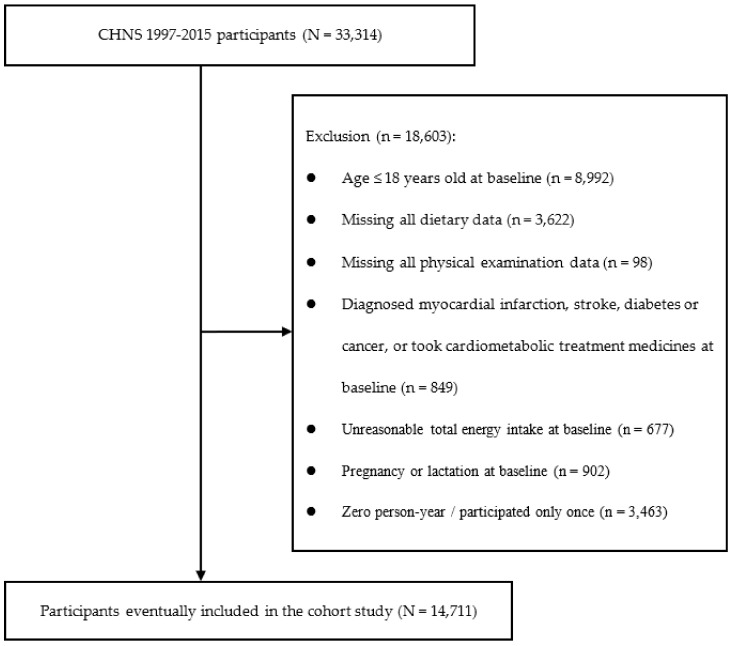
Flow chart of the prospective study with 14,711 Chinese adults from the China Health and Nutrition Survey (CHNS), 1997–2015.

**Figure 2 nutrients-14-02055-f002:**
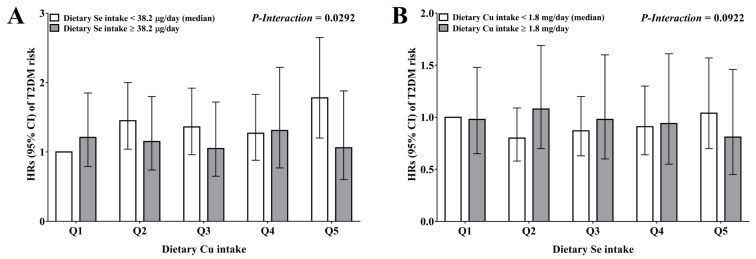
Joint analysis of dietary Cu and Se intake on T2DM risk. (**A**) The joint analysis of dietary Cu (quintiles) and Se (median) on T2DM risk. (**B**) The joint analysis of dietary Se (quintiles) and Cu (median) on T2DM risk. Data was presented as HR (95% CI) estimated by using time-dependent Cox proportional hazard regression models. Models adjusted for sex (women or men), age (<50, 50–54, 55–59, 60–64 or ≥65 years), BMI (<18.5, 18.5–23.9, 24–27.9 or ≥28 kg/m^2^), urban index (tertiles), region [north (Beijing, Liaoning, Heilongjiang, Shandong, and Henan) or south (Shanghai, Jiangsu, Hubei, Hunan, Guangxi, Guizhou, and Chongqing)], education level (primary school, middle school, or high school), alcohol intake (ever drinking or not), smoking (ever smoking or not), physical activity (tertiles), baseline hypertension (yes or no), and dietary variables, including total energy intake, dietary GI, fiber intake, animal to plant protein ratio, PUFA to SFA ratio, and cholesterol intake (all quintiles). Abbreviations: Cu, copper; GI, glycemic index; PUFA, polyunsaturated fatty acid; Q, quintiles; Se, Selenium; SFA, saturated fatty acid; T2DM, type 2 diabetes mellitus.

**Table 1 nutrients-14-02055-t001:** Baseline characteristics of 14,711 Chinese adults in the China Health and Nutrition Survey, 1997–2015, according to quintiles of dietary Cu and Se intakes ^1^.

Variables	All	Quintiles of Dietary Cu Intakes			Quintiles of Dietary Se Intakes		
		Q1	Q3	Q5	Q1	Q3	Q5
N	14,711	2942	2945	2942	2943	2943	2942
Cu, mg/day	1.9 ± 0.6	1.3 ± 0.2	1.8 ± 0.1	2.7 ± 0.9	1.8 ± 0.5	1.8 ± 0.5	2.0 ± 0.9
Se, μg/day	41.1 ± 17.6	38.8 ± 14.0	40.6 ± 16.6	44.8 ± 22.8	23.9 ± 4.5	38.2 ± 1.7	65.4 ± 22.9
Age, years	45 ± 15	47 ± 15	44 ± 15	44 ± 15	47 ± 16	44 ± 14	45 ± 15
Male, *n* (%)	7333 (49.8)	1534 (52.1)	1434 (48.7)	1438 (48.9)	1363 (46.3)	1504 (51.1)	1578 (53.6)
BMI, kg/m^2^	23.2 ± 3.3	23.5 ± 3.4	23.1 ± 3.4	23.3 ± 3.3	22.5 ± 3.4	23.3 ± 3.2	23.9 ± 3.5
Urbanization index, *n* (%)							
Low	4902 (33.3)	553 (18.8)	1059 (36.0)	1229 (41.8)	1581 (53.7)	940 (31.9)	467 (15.9)
Medium	4903 (33.3)	1031 (35.0)	1032 (35.0)	788 (26.8)	876 (29.8)	1068 (36.3)	869 (29.5)
High	4906 (33.4)	1358 (46.2)	854 (29.0)	925 (31.4)	486 (16.5)	935 (31.8)	1606 (54.6)
Region, *n* (%)							
Northern	6112 (41.6)	917 (31.2)	1198 (40.7)	1589 (54.0)	946 (32.1)	1118 (38.0)	1630 (55.4)
Southern	8589 (58.4)	2025 (68.8)	1747 (59.3)	1353 (46.0)	1997 (67.9)	1825 (62.0)	1312 (44.6)
Education level, *n* (%)							
Primary	6883 (46.8)	1075 (36.5)	1466 (49.8)	1457 (49.5)	1827 (62.1)	1342 (45.6)	917 (31.2)
Middle	4188 (28.5)	908 (30.9)	848 (28.8)	796 (27.1)	733 (24.9)	885 (30.1)	884 (30.0)
High	3640 (24.7)	959 (32.6)	631 (21.4)	689 (23.4)	383 (13.0)	716 (24.3)	1141 (38.8)
Alcohol intake, *n* (%)							
No	7647 (52.0)	1618 (55.0)	1485 (50.4)	1556 (52.9)	1695 (57.6)	1469 (49.9)	1463 (49.7)
Yes	7064 (48.0)	1324 (45.0)	1460 (49.6)	1386 (47.1)	1248 (42.4)	1474 (50.1)	1479 (50.3)
Smoking status, *n* (%)							
No	9314 (63.3)	1906 (64.8)	1842 (62.5)	1916 (65.1)	1875 (63.7)	1829 (62.1)	1898 (64.5)
Yes	5397 (36.7)	1036 (35.2)	1103 (37.5)	1026 (34.9)	1068 (36.3)	1114 (37.9)	1044 (35.5)
Physical activity status, METs-h/week	103.2 ± 83.8	88.3 ± 77.0	106.2 ± 82.6	111.4 ± 90.5	126.0 ± 90.6	100.0 ± 82.5	82.1 ± 73.9
Hypertension, *n* (%)							
No	11610 (78.9)	2271 (77.2)	2364 (80.3)	2283 (77.6)	2345 (79.7)	2367 (80.4)	2254 (76.6)
Yes	3101 (21.1)	671 (22.8)	581 (19.7)	659 (22.4)	598 (20.3)	576 (19.6)	688 (23.4)
Total energy, kcal/day	2104.6 ± 136.5	2027.5 ± 210.8	2125.9 ±95.8	2132.0 ± 88.1	2054.2 ± 200.7	2118.2 ± 113.6	2121.7 ± 104.7
Protein, % energy	12.3 ± 2.7	11.8 ± 3.5	12.0 ± 2.1	13.2 ± 2.8	10.5 ± 3.0	12.1 ± 1.7	14.5 ± 2.7
Animal protein, % energy	3.8 ± 2.7	4.5 ± 2.6	3.7 ± 2.4	3.4 ± 3.1	1.9 ± 1.5	3.8 ± 2.0	6.0 ± 3.2
Plant protein, % energy	7.7 ± 2.0	6.1 ± 1.8	7.8 ± 1.4	9.1 ± 2.2	7.8 ± 2.1	7.7 ± 1.9	7.5 ± 2.1
Animal protein: plant protein ratio	0.6 ± 3.7	1.0 ± 8.2	0.5 ± 0.5	0.5 ± 0.6	0.5 ± 8.2	0.6 ± 1.0	1.0 ± 0.8
Fat, % energy	31.7 ± 10.5	40.0 ± 11.4	30.2 ± 8.2	26.7 ± 9.9	29.2 ± 12.8	32.3 ± 9.8	33.6 ± 9.1
SFA, % energy	7.3 ± 3.0	9.6 ± 3.4	7.0 ± 2.3	5.8 ± 2.6	6.6 ± 3.5	7.5 ± 2.9	8.0 ± 2.6
MUFA, % energy	12.6 ± 5.3	16.8 ± 5.9	12.1 ± 4.1	9.8 ± 4.6	11.7 ± 6.3	12.9 ± 5.1	13.2 ± 4.6
PUFA, % energy	8.2 ± 4.2	9.4 ± 5.5	7.7 ± 3.6	7.8 ± 3.7	7.8 ± 5.3	8.3 ± 3.9	8.3 ± 3.6
PUFA: SFA ratio	1.3 ± 0.7	1.1 ± 0.6	1.2 ± 0.6	1.5 ± 0.7	1.4 ± 0.9	1.3 ± 0.6	1.1 ± 0.5
Cholesterol, % energy	0.07 ± 0.06	0.07 ± 0.06	0.06 ± 0.05	0.06 ± 0.06	0.04 ± 0.04	0.06 ± 0.05	0.10 ± 0.08
Carbohydrate, % energy	54.3 ± 11.0	46.4 ± 11.0	56.1 ± 9.2	58.2 ± 11.2	58.7 ± 12.4	54.0 ± 10.0	49.8 ± 9.9
Fiber, g/day	10.4 ± 5.0	7.2 ± 3.3	10.2 ± 4.1	13.9 ± 6.3	9.6 ± 4.8	10.0 ± 4.3	11.6 ± 6.5
Dietary GI	69.0 ± 6.7	68.0 ± 7.8	70.0 ± 5.6	68.0 ± 7.3	68.6 ± 7.0	69.3 ± 6.2	68.2 ± 7.4

^1^ Amounts of nutrient intakes were adjusted for total energy. Data were presented as mean ± SD or *n* (%). Abbreviations: Cu, copper; FA, fatty acid; GI, glycemic index; METs-h: metabolic equivalent task hour; MUFA, monounsaturated fatty acid; PUFA, polyunsaturated fatty acid; Q, quintiles; SD, standard deviation; Se, Selenium; SFA, saturated fatty acid.

**Table 2 nutrients-14-02055-t002:** Major food sources of dietary Cu and Se intakes in 14,711 Chinese adults of the China Health and Nutrition Survey, 1997–2015 ^1^.

Nutrients	Food Sources	Median (IQR)
Dietary Cu intakes, mg/day	Grains and tubers	0.98 (0.67, 1.32)
	Vegetables, fruits, fungi, and algae	0.23 (0.15, 0.35)
	Legumes	0.15 (0.05, 0.30)
	Meat	0.06 (0.02, 0.12)
	Ethnic foods, cakes, and fast food	0.06 (0.02, 0.12)
	Condiments	0.04 (0.01, 0.04)
	Eggs	0.02 (0.01, 0.04)
	Fish and seafoods	0.01 (0, 0.02)
	Others	0 (0, 0.03)
Dietary Se intakes, µg/day	Grains and tubers	14.14 (9.66, 21.39)
	Meat	7.52 (3.20, 12.90)
	Eggs	2.73 (0.80, 5.26)
	Vegetables, fruits, fungi, and algae	2.05 (1.36, 2.96)
	Fish and seafoods	1.29 (0, 5.94)
	Legumes	0.87 (0.28, 1.76)
	Ethnic foods, cakes, and fast food	0.51 (0.07, 1.52)
	Condiments	0.47 (0.24, 0.88)
	Others	0 (0, 0.13)

^1^ Data were presented as median (IQR). Abbreviations: Cu, copper; IQR, interquartile range; SFA, saturated fatty acid.

**Table 3 nutrients-14-02055-t003:** HRs and 95% CIs for T2DM risk associated with quintiles of dietary Cu and Se intakes in 14,711 Chinese adults of the China Health and Nutrition Survey, 1997–2015 ^1^.

Variables	Q1	Q2	Q3	Q4	Q5	*p*-Trend
Dietary Cu intakes, mg/day						
Range	0–1.51	1.51–1.73	1.73–1.93	1.93–2.21	2.21–19.66	
Median	1.34	1.63	1.83	2.05	2.46	
Cases (rate, ‰)	133 (6.44)	219 (7.24)	217 (6.42)	238 (6.99)	233 (8.20)	
Model 1	1.00 (Ref)	0.90 (0.71, 1.13)	0.73 (0.56, 0.97)	0.79 (0.55, 1.11)	0.95 (0.62, 1.47)	0.17
Model 2	1.00 (Ref)	1.19 (0.93, 1.52)	1.11 (0.81, 1.53)	1.22 (0.80, 1.85)	1.30 (0.77, 2.20)	0.0130
Dietary Se intakes, µg/day						
Range	0–29.31	29.31–35.33	35.33–41.35	41.35–50.56	50.56–433.16	
Median	25.08	32.36	38.16	45.36	59.42	
Cases (rate, ‰)	190 (6.70)	204 (6.29)	214 (6.66)	233 (7.89)	199 (8.07)	
Model 1	1.00 (Ref)	1.01 (0.81, 1.27)	1.26 (0.95, 1.67)	1.68 (1.17, 2.41)	1.94 (1.25, 2.99)	<0.0001
Model 2	1.00 (Ref)	0.95 (0.75, 1.20)	0.93 (0.68, 1.26)	0.92 (0.62, 1.37)	0.89 (0.54, 1.45)	0.82

^1^ Data was presented as HR (95% CI) estimated by using time-dependent Cox proportional hazard regression models. Rate was calculated as incidence density. Model 1 adjusted for sex (women or men) and age (<50, 50–54, 55–59, 60–64 or ≥65 years). Model 2 additionally adjusted for BMI (<18.5, 18.5–23.9, 24–27.9 or ≥28 kg/m^2^), urban index (tertiles), region [north (Beijing, Liaoning, Heilongjiang, Shandong, and Henan) or south (Shanghai, Jiangsu, Hubei, Hunan, Guangxi, Guizhou, and Chongqing)], education level (primary school, middle school, or high school), alcohol intake (ever drinking or not), smoking (ever smoking or not), physical activity (tertiles), baseline hypertension (yes or no), and dietary variables, including total energy intake, dietary GI, fiber intake, animal to plant protein ratio, PUFA to SFA ratio, and cholesterol intake (all quintiles). Abbreviations: Cu, copper; GI, glycemic index; PUFA, polyunsaturated fatty acid; Q, quintiles; Se, Selenium; SFA, saturated fatty acid; T2DM, type 2 diabetes mellitus.

**Table 4 nutrients-14-02055-t004:** HRs and 95% CIs for T2DM risk associated with dietary Cu and Se intakes in 14,711 Chinese adults of the China Health and Nutrition Survey, 1997–2015, stratified by age, sex, BMI, and baseline hypertension history ^1^.

Variables	N	Q1	Q2	Q3	Q4	Q5	*p*-Trend	*p*-Interaction
Dietary Cu intakes, mg/day								
Age, years								
<60	12,004	1.00 (Ref)	1.11 (0.83, 1.48)	1.05 (0.73, 1.53)	1.11 (0.68, 1.82)	1.17 (0.63, 2.17)	0.0281	0.91
≥60	2707	1.00 (Ref)	1.09 (0.68, 1.76)	0.82 (0.44, 1.52)	0.97 (0.44, 2.13)	0.99 (0.37, 2.62)	0.34	
Sex								
Male	7333	1.00 (Ref)	1.33 (0.94, 1.89)	0.95 (0.59, 1.51)	0.93 (0.50, 1.71)	0.90 (0.41, 1.95)	0.15	0.20
Female	7378	1.00 (Ref)	1.04 (0.73, 1.48)	1.25 (0.80, 1.95)	1.52 (0.86, 2.69)	1.68 (0.82, 3.43)	0.0236	
BMI, kg/m^2^								
<24.0	9403	1.00 (Ref)	1.43 (0.96, 2.15)	1.16 (0.68, 1.98)	1.52 (0.76, 3.05)	1.56 (0.66, 3.72)	0.0119	0.14
≥24.0	5308	1.00 (Ref)	1.01 (0.73, 1.38)	1.09 (0.73, 1.64)	1.06 (0.62, 1.80)	1.19 (0.61, 2.32)	0.21	
Region								
North	6211	1.00 (Ref)	0.96 (0.65, 1.41)	0.83 (0.53, 1.30)	0.94 (0.53, 1.65)	0.81 (0.40, 1.65)	0.29	0.36
South	8589	1.00 (Ref)	1.46 (1.05, 2.03)	1.51 (0.96, 2.40)	1.56 (0.84, 2.90)	2.17 (1.00, 4.69)	0.0180	
Hypertension								
No	11,610	1.00 (Ref)	1.20 (0.87, 1.66)	1.24 (0.81, 1.90)	1.39 (0.80, 2.42)	1.55 (0.77, 3.12)	0.0077	0.0347
Yes	3101	1.00 (Ref)	1.14 (0.77, 1.69)	0.96 (0.58, 1.60)	1.06 (0.55, 2.04)	1.08 (0.48, 2.41)	0.55	
Dietary Se intakes, µg/day								
Age, years								
<60	12,004	1.00 (Ref)	0.92 (0.70, 1.21)	0.78 (0.54, 1.13)	0.80 (0.50, 1.29)	0.76 (0.42, 1.37)	0.41	0.08
≥60	2707	1.00 (Ref)	1.02 (0.63, 1.64)	1.44 (0.78, 2.64)	1.19 (0.55, 2.55)	1.11 (0.43, 2.83)	0.67	
Sex								
Male	7333	1.00 (Ref)	1.18 (0.82, 1.70)	0.92 (0.57, 1.50)	1.21 (0.65, 2.23)	1.03 (0.48, 2.19)	0.99	0.22
Female	7378	1.00 (Ref)	0.81 (0.59, 1.11)	0.99 (0.66, 1.49)	0.75 (0.44, 1.27)	0.86 (0.45, 1.65)	0.87	
BMI, kg/m^2^								
<24.0	9403	1.00 (Ref)	1.02 (0.72, 1.45)	0.85 (0.52, 1.39)	0.88 (0.46, 1.66)	0.87 (0.39, 1.90)	0.91	0.18
≥24.0	5308	1.00 (Ref)	0.89 (0.65, 1.23)	1.01 (0.67, 1.51)	1.07 (0.64, 1.79)	1.03 (0.55, 1.93)	0.78	
Region								
North	6211	1.00 (Ref)	0.83 (0.58, 1.17)	0.77 (0.50, 1.18)	0.61 (0.36, 1.06)	0.57 (0.29, 1.13)	0.37	0.24
South	8589	1.00 (Ref)	1.00 (0.72, 1.40)	1.02 (0.63, 1.64)	1.31 (0.70, 2.43)	1.21 (0.57, 2.59)	0.78	
Hypertension								
No	11,610	1.00 (Ref)	0.98 (0.72, 1.32)	0.91 (0.61, 1.38)	0.89 (0.52, 1.52)	0.70 (0.36, 1.37)	0.18	0.72
Yes	3101	1.00 (Ref)	0.81 (0.54, 1.19)	0.81 (0.49, 1.32)	0.83 (0.45, 1.52)	1.03 (0.50, 2.13)	0.19	

^1^ Data was presented as HR (95% CI) estimated by using time-dependent Cox proportional hazard regression models. Models adjusted for sex (women or men), age (<50, 50–54, 55–59, 60–64 or ≥65 years), BMI (<18.5, 18.5–23.9, 24–27.9 or ≥28 kg/m^2^), urban index (tertiles), region [north (Beijing, Liaoning, Heilongjiang, Shandong, and Henan) or south (Shanghai, Jiangsu, Hubei, Hunan, Guangxi, Guizhou, and Chongqing)], education level (primary school, middle school, or high school), alcohol intake (ever drinking or not), smoking (ever smoking or not), physical activity (tertiles), baseline hypertension (yes or no), and dietary variables, including total energy intake, dietary GI, fiber intake, animal to plant protein ratio, PUFA to SFA ratio, and cholesterol intake (all quintiles). Abbreviations: Cu, copper; GI, glycemic index; PUFA, polyunsaturated fatty acid; Q, quintiles; Se, Selenium; SFA, saturated fatty acid; T2DM, type 2 diabetes mellitus.

## Data Availability

Not applicable.

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
