# Peer review of "Dietary Copper and Selenium Intakes and the Risk of Type 2 Diabetes Mellitus: Findings from the China Health and Nutrition Survey"

_nutrients, 2022, doi:10.3390/nu14102055_

Round 1

Reviewer 1 Report

I have found this paper well done and interesting.

I appreciated the deep discussion and the correct references to microbiota studies in the field.

I believe methods and results are thouroughly described.

English language is fine and only few, minor mistakes deserve correction.

Stats are employed in the right way.

Square and round brackets use is correct (a rare thing).

References are sufficiently updated.

The only concern I have is related to the absent discussion about what food, in the Chinese kitchen, contain copper and selenium. I believe a single table will explain chinese food content of both copper and selenium and help the reader agree with the results presented in this study.

Finally the sentence in the abstract (rows 23, 24): "dietary Cu and Se intakes were associated with decreased or increased T2DM risk, respectively" is cumbersome and, may be, confusing. I think something like: "...dietary Cu intake was associated with decreased T2DM risk, while Se with increased T2DM risk" would be clearer.

Author Response

Reviewer 1: I have found this paper well done and interesting. I appreciated the deep discussion and the correct references to microbiota studies in the field. I believe methods and results are thouroughly described. English language is fine and only few, minor mistakes deserve correction. Stats are employed in the right way. Square and round brackets use is correct (a rare thing). References are sufficiently updated.

Response: We thank the reviewer for the encouragement. The manuscript has been thoroughly revised due to a mistake in the first version. When we explored the food sources of dietary Cu and Se, we found that we missed cooking oils and condiments in the prior food and nutrient intake calculation because these foods were only measured in the food-weighing methods but not in 3 day 24h dietary recalls. We recalculated food and nutrient intakes following inclusion of cooking oils and condiments. This recalculation resulted in alterations in total energy intakes of participants, which was related to one of our exclusion criteria, and the final sample size was also changed as a consequence. Following reanalysis, dietary Cu or Se intake was no longer significantly associated with T2DM risk. Fortunately, the interaction between dietary Cu and Se intakes on T2DM still existed, and dietary Se intake significantly modified the association between dietary Cu intake and T2DM risk. 

1) The only concern I have is related to the absent discussion about what food, in the Chinese kitchen, contain copper and selenium. I believe a single table will explain chinese food content of both copper and selenium and help the reader agree with the results presented in this study.

Response: We thank the reviewer for pointing out this question. Consistent with the reviewer’s request, food sources of dietary Cu and Se intakes of participants were categorized into “grains and tubers”, “legumes”, “vegetables, fruits, fungi and algae”, “meat”, “eggs”, “fish and seafoods”, “condiments", “ethnic foods and cakes and fast food” and “others” on the basis of our data. This information has now been provided on Page 4 Lines 122 to 126. Data are presented on Page 9 Lines 259-262 and Table 2.

2) Finally, the sentence in the abstract (rows 23, 24): "dietary Cu and Se intakes were associated with decreased or increased T2DM risk, respectively" is cumbersome and, may be, confusing. I think something like: "...dietary Cu intake was associated with decreased T2DM risk, while Se with increased T2DM risk" would be clearer.

Response: We thank the reviewer for pointing out this issue. This sentence has now been revised as “dietary Cu or Se intake independently was not associated with decreased T2DM risk” Please see Page 1 Lines 27-28 for revisions.

Reviewer 2 Report

Reviewer comments and suggestions

The study examined prospective associations between dietary Cu and Se intakes and T2DM risk in Chinese adults. The data was taken from China Health and Nutrition Survey (1997-2015) which recruited 14,440 adults. The studied nutrient intakes were measured by 3 consecutive 24-h recalls. T2DM was diagnosed by a validated questionnaire and laboratory examination. A total of 1,061 T2DM cases were diagnosed. In fully-adjusted models, the HRs (95% CIs) for T2DM in Q5 of Cu and Se intakes were 0.60 (0.38, 0.95) and 2.07 (1.22, 3.53) compared with Q1, respectively (both P-trends <0.05). Cu and Se intakes significantly interacted with each other on T2DM risk (both P-interaction<0.05). In this study, dietary Cu and Se intakes were associated with decreased or increased T2DM risk in Chinese adults. 

The paper has nicely complied and covered. However, in many places, the authors need to do a few small corrections to the manuscript. Based on my view, below are the comments that need to be incorporated into the revised version of the manuscript. 

  1. Line 23-26; The result would be clearly stated in line, I am not convinced with increase or decrease, the sentence need to be modified. Moreover, the merging of the sentence is needed.
  2. Line 34-35 the sentence needs to be updated with information on certain micronutrients"
  3. Line 52 Animal studies have shown that both selenium deficiency and (the lines need 2-3 relevant references) 
  4. Line 56-57 reference 17 and 18 please explore these studies
  5. Line 74-75 Ethical number should be important here, please indicate
  6. Line 104-106 please mention all the information here
  7. You need to mention in the study design if it is possible. It is advised to elaborate the study design in figure 1.
  8. Line 167-168 please mention when you collected baseline and the data of follow up
  9. In line 204 again the author missed the points, better to indicate the time period of baseline and follow up
  10. Line 287-288 repeated lines
  11. Line 301-303 The lines need to be confused
  12. Line 318-321 avoid long sentence
  13. Line 334 could you please link gut microbiota with se
  14. Please check reference 25 and in some references of MDPI journal such as nutrients, the author did not write the page number. It should be consistent. Please check the issues. 

Author Response

Reviewer 2: The paper has nicely complied and covered. However, in many places, the authors need to do a few small corrections to the manuscript. Based on my view, below are the comments that need to be incorporated into the revised version of the manuscript.

Response: We thank the reviewer for the comments. The manuscript has been thoroughly revised due to a mistake in the first version. When we explored the food sources of dietary Cu and Se, we found that we missed cooking oils and condiments in the prior food and nutrient intake calculation because these foods were only measured in the food-weighing methods but not in 3 day 24h dietary recalls. We recalculated food and nutrient intakes following inclusion of cooking oils and condiments. This recalculation resulted in alterations in total energy intakes of participants, which was related to one of our exclusion criteria, and the final sample size was also changed as a consequence. Following reanalysis, dietary Cu or Se intake was no longer significantly associated with T2DM risk. Fortunately, the interaction between dietary Cu and Se intakes on T2DM still existed, and dietary Se intake significantly modified the association between dietary Cu intake and T2DM risk.

1) Line 23-26; The result would be clearly stated in line, I am not convinced with increase or decrease, the sentence needs to be modified. Moreover, the merging of the sentence is needed.

Response: We thank the reviewer for pointing out this issue. This sentence has now been revised as “dietary Cu or Se intake independently was not associated with T2DM risk” Please see Page 1 Lines 27-28 for revisions.

2) Line 34-35 the sentence needs to be updated with information on certain micronutrients"

Response: Consistent with the reviewer’s request, we have specified the certain micronutrients as selenium (Se), copper (Cu), calcium, iron, manganese and magnesium, and one more reference has been added to support the point. Please see Page 1 Line 40 for revisions.

3) Line 52 Animal studies have shown that both selenium deficiency and (the lines need 2-3 relevant references)

Response: We thank the reviewer for pointing out this omission. In response to the reviewer’s request, reference 13 to 15 have now been added to support the point.

4) Line 56-57 reference 17 and 18 please explore these studies

Response: In response to the reviewer’s suggestion, the results of these two studies have been explored in details on Page 2 Lines 62-67.

5) Line 74-75 Ethical number should be important here, please indicate

Response: We thank the reviewer for pointing out this omission. Consistent with the reviewer’s request, we have been added the ethical numbers on Page 2 Lines 84 to 86.

6) Line 104-106 please mention all the information here

Response: In response to the reviewer’s suggestion, the “nutrients intake” has been specified as “intakes of dietary total energy, Cu, Se, total protein, animal protein, plant protein, fat, saturated fatty acids (SFA), monounsaturated fatty acids (MUFA), polyunsaturated fatty acids (PUFA), cholesterol, carbohydrate and fiber” on Pages 3-4 Lines 116 to 118.

7) You need to mention in the study design if it is possible. It is advised to elaborate the study design in figure 1.

Response: We thank the reviewer for pointing out this issue. The study design of CHNS study has been described briefly on Page 2 Lines 75-87. In terms of the study design of the current investigation, we have presented the inclusion and exclusion process on Pages 2-3 Lines 90-101, and have elaborated the study design as a prospective cohort study on Page 2 Line 88 and Figure 1 Page 3 Line 102 in response to the reviewer’s request.

8) Line 167-168 please mention when you collected baseline and the data of follow up.

Response: We thank the reviewer for the suggestion. The baseline time of each participant was defined as the date of their first participation with complete dietary data since 1997, and the individual follow-up person-years of each participant were calculated from the baseline to the date of first diagnosis with T2DM, death, the last wave before departure from the survey or the end of the last survey at 2015 whichever came first. This information has been specified on Page 5 Lines 188 to 193 for clarification. Reference 37 was been added to further support the point.

9) In line 204 again the author missed the points, better to indicate the time period of baseline and follow up.

Response: We thank the reviewer for pointing out this omission. We have now added the time period of baseline and follow-up, which was duration of 9.2 years (147,142 person-years), on Page 6 lines 226-228.

10) Line 287-288 repeated lines.

Response: We thank the reviewer for pointing out this issue. The repeated lines have now been deleted.

11) Line 301-303 The lines need to be confused.

Response: We thank the reviewer for pointing out this issue. The confused sentences have now been deleted.

12) Line 318-321 avoid long sentence.

Response: In response to the reviewer’s concern, the sentence has been trimmed and now read as “potential mechanisms underlying the positive association may be partially attributed to the adverse effects of high doses of Se compounds and species or subsequent upregulation of selenoprotein biosynthesis on glucose homeostasis”. Please see Page 15 Lines 375-378 for details.

13) Line 334 could you please link gut microbiota with se.

Response: We thank the reviewer for the suggestion. The association between dietary Se intake and gut microbiota in human study has been provided on Page 16 Lines 388-391. In response to the reviewer’s suggestion, we have further added some results from an animal study which links gut microbiota and Se. Please see Page 16 Lines 391-393 for details.

14) Please check reference 25 and in some references of MDPI journal such as nutrients, the author did not write the page number. It should be consistent. Please check the issues.

Response: We thank the reviewer for pointing out this issue. All the page numbers of references have been added for consistency.
